# Effect of Cross-Linker Length on the Absorption Characteristics of the Sodium Salt of Cross-Linked Polyaspartic Acid

**DOI:** 10.3390/polym14112244

**Published:** 2022-05-31

**Authors:** Junchao Xu, Kyu Oh Kim, Kee Jong Yoon

**Affiliations:** 1Jiangsu New Vision Advanced Functional Fiber Innovation Center Co., Ltd., Suzhou 215000, China; xujunchao0306@gmail.com; 2Department of Fiber System Engineering, Dankook University, 152, Jookjeon-ro, Suji-gu, Yongin-si 16890, Korea; affablekim@gmail.com

**Keywords:** superabsorbent polymer, biodegradable, polysuccinimide, polyaspartic acid Na salt, cross-linking length

## Abstract

For the development of biodegradable superabsorbent polymers, the effect of the cross-linking length on the absorption characteristics of the Na salt of polyaspartic acid (PAspNa) was demonstrated using different concentrations of diamine cross-linking agents bearing carbon chains of different lengths, viz., ethylenediamine, 1,6-hexamethylenediamine, 1,8-diaminooctane, 1,10-diaminodecane, and 1,12-diaminododecane were used as cross-linking agents. The absorption of PAspNa was measured in deionised water and in a 0.9% aqueous NaCl solution. Under the conditions tested, when the alkyl chain of PAspNa was too short or too long, the absorbency was low and the cross-linking length was optimum. The success of the cross-linking reaction was confirmed by FT-IR spectroscopy. The degree of cross-linking was estimated and the ideal concentration for maximum water absorption was determined by elemental analysis. The sample obtained by cross-linking 1,8-diaminooctane at a concentration of 0.11 g/g polysuccinimide (PSI) showed the highest absorption. The thermal properties of each material were determined by dynamic scanning calorimetry. Therefore, the length of the cross-linking agent was found to strongly influence water absorption.

## 1. Introduction

Superabsorbent polymers (SAPs) are functional hydrophilic polymer materials with three-dimensional network structures, bearing many strongly hydrophilic groups, such as carboxyl, hydroxyl, and amino groups, in the molecular chain, and having a constant cross-linking density [1,2,3,4]. SAPs absorb water in quantities hundreds to thousands of times their own weight, have high water-holding capacities, and are not easily dehydrated even under external pressure, thus implying robust gel strengths. However, current widely used SAPs cause environmental pollution because they are not biodegradable. Therefore, research and development of biodegradable SAPs is required [5].

Currently, the most widely used superabsorbent is sodium polyacrylate, which has excellent water absorption properties and a simple synthesis process but is not biodegradable. Biodegradable superabsorbents can be developed using starch, polysaccharides, sodium alginate, and natural amino acid polymers as raw materials, but they have the disadvantages of a high cost and low water absorption properties. Biodegradable superabsorbents have excellent salt tolerance and hydrophilicity, as well as low production costs. Such comprehensive characteristics are the primary directions for future research.

The biodegradability of superabsorbents is related to the structure of the constituent polymer. Biodegradation is possible if the main chain of the resin contains amide, ester, urethane, urea, and glucosidic bonds [6]. Polysaccharides, such as starch, cellulose, and chitosan, are generally biodegradable because their molecules contain glycosidic bonds. However, the main chain of the currently used superabsorbent is composed of C–C bonds, which are difficult to decompose. Polyacrylic acid sodium salt, which is mainly used as a superabsorbent, shows good liquid absorption, but is not decomposed by the action of microorganisms and has poor degradability; therefore, it may cause environmental pollution when discarded after use.

Starch can be obtained from a variety of sources and is thus inexpensive. In starch-based superabsorbents, which are resins mainly prepared by graft copolymerisation of an acrylic monomer with starch, starch is biodegraded by the action of microorganisms, but the graft chain is difficult to decompose and thus provides superabsorbency [7,8,9,10]. Common acrylic monomers include acrylic acid, acrylamide, and acrylonitrile. In terms of application, starch-based superabsorbents are extremely good water absorbents, and their biodegradability has been extensively studied. Liu et al. investigated the biodegradability of starch-grafted acrylic acid absorbent resin and reported that the resin could be degraded via the decomposition of both the starch backbone and the polyacrylic acid graft molecules by microorganisms [11]. Lanthong et al. reported the synthesis of α-amylase using a redox initiator system containing ammonium sulfate and *N*,*N*,*N*′,*N*′-tetramethyl ethylene diamine for designing acrylamide/itaconic acid–tapioca starch superabsorbent. It was shown that hydrolysis occurred and converted the groups into glucose units, which showed excellent degradability [12]. Jin et al. reported that a starch-g-(acrylic acid-co-acrylamide) absorbent resin was prepared and after 55 days of being kept buried in soil, 32% of the resin was decomposed in the absorbent resin containing 20% starch [13]. Polyamino acids are polymers of amino acids in which the main chains are linked by peptide bonds. The side chain contains some reactive groups, such as –COOH and –CONH_2_. Polyamino acid absorbent resin is a new type of biodegradable absorbent resin. Currently, the amino acids that can reportedly be used to prepare this absorbent resin are polyaspartic acid (PAsp), polylysine, and polyglutamic acid. Using succinimide as a raw material and hexamethylene diamine and butanediamine as the cross-linking agents, Fang et al. developed a two-step cross-linking method for synthesising a PAsp superabsorbent that absorbed distilled water and physiological saline [14]. The absorbency of the distilled water solution was 1250 g/g, the absorbency of the salt solution was 147 g/g, and the gel strength was 15.02 s. Both first- and second-stage resins showed excellent biodegradability. Wei et al. synthesised a PAsp superabsorbent using a simultaneous heterogeneous hydrolysis/cross-linking reaction, and the maximum absorption obtained was 593 g/g [15]. It was confirmed that the main chain of the resin was cleaved and decomposed in an enzyme decomposition experiment. The starch and fibre superabsorbents studied were degraded only partially. Although there have been many studies on PAsp, there has been little research on its biodegradability.

The Na salt of polyaspartic acid (PAspNa) is a polyamino acid polymer material that can absorb large amounts of water, has good biodegradability, is nontoxic, and contains large amounts of amino and carboxyl groups. Therefore, PAsp is an ecofriendly product. In addition, PAsp has excellent chelating properties and dispersibility, and thus, has widespread applications and great market demand. The monomer of the degradable, water-soluble polymer PAsp is L-aspartic acid (L-asp), which can have two types of amide bonds [16].

Depending on the starting material, PAsp synthesis methods are primarily classified as either L-asp-based or maleic-anhydride-based methods. Among them, the L-asp-based method uses phosphoric acid as a catalyst at a temperature of ~200 °C to form polysuccinimide (PSI) by thermal condensation of L-asp monomers in vacuum [17]. After purification, filtration cross-linking reaction, and alkaline hydrolysis, a PAsp superabsorbent is obtained.

In this study, the optimal conditions for water absorption by PAspNa in terms of the chain length of the diamine cross-linking agent were studied. In the future, we plan to study their biodegradability. First, high-molecular-weight PSI was prepared using L-asp as a raw material and then cross-linked at 40 °C. In this study, a method for changing the length and concentration of the cross-linking agent diamine to obtain the best absorption under constant reaction conditions was explored. Success of the cross-linking reaction was confirmed using Fourier transform infrared (FT-IR) spectroscopy, and the structure of the superabsorbent was determined. Elemental analysis was used to determine the actual amount of cross-linking agent and to obtain basic data for better absorption of PAsp. The thermal properties of each material were obtained using differential scanning calorimetry (DSC). The biodegradable and biofriendly superabsorbent of cross-linked PAspNa argues in favour of the feasibility of future biological applications.

## 2. Experiments and Method

### 2.1. Materials and Reagents

The L-asp used in this study was obtained from Hebei Think-Do Environment Co., Ltd., Shijiazhuang, China, and was used without further purification. *N*,*N*-dimethylformamide (DMF, Samchun, Seoul, Korea, 99%), 85% phosphoric acid (Samchun, 85%), methanol (Duksan Pure Chemicals Co. Ltd., Ansan-si, Korea), and sodium hydroxide (Samchun, 98%) were used without purification. Ethylenediamine (EDA, Sigma-Aldrich, Burlington, MA, USA), 1,6-hexamethylene diamine (HMDA Sigma-Aldrich), 1,8-diaminooctane (Tokyo Chemical Industry Co. Ltd., Tokyo, Japan), 1,10-diaminodecane (Sigma-Aldrich), and 1,12-diaminododecane (Sigma-Aldrich) were used as cross-linking agents.

### 2.2. Synthesis of PSI

After preheating 100 g of L-Asp at 80 °C for 1 h, 10 mL of 85% H_3_PO_4_ was added as a catalyst. The temperature was raised to 200 °C with stirring, and the resultant product was thermally polymerised under reduced pressure in a 0.095 MPa vacuum for 2 h. After pulverising the product with a grinder, 10 mL of 85% H_3_PO_4_ was added, stirred, and the second thermal polymerisation was carried out in a vacuum for 2 h. The product was pulverised again in a grinder, 5 mL of 85% H_3_PO_4_ was added, stirred, and the third thermal polymerisation was performed in vacuum for 2 h. The pulverised PSI product was dissolved in excess DMF and was purified by precipitation with H_2_O. This was again washed with H_2_O to neutrality, filtered, and dried at 50 °C to obtain PSI [18].

### 2.3. Synthesis of PAspNa by Cross-Linking and Hydrolysis of PSI

Briefly, 40 mL of DMF solvent and 2 g of PSI were added to a 100 mL beaker, stirred, and 20 mL of deionised water was added as a dispersant. The mixture comprising PSI, DMF, and deionised water was stirred for 0.5 h. Then, 0.11 g of a cross-linking agent was added to the beaker and the reaction was continued for 1 h at 40 °C to afford cross-linked PSI. Next, NaOH was added until the pH was 9 to hydrolyse the imide ring of the cross-linked polymer, followed by hydrolysis at 40 °C. CH_3_OH (50 mL) was added to the precipitate, followed by filtration and vacuum drying at 40 °C.

### 2.4. Elemental Analysis

An element (C, N, S) analyser (Flash EA 1112, Thermo Electron Corporation, Waltham, MA, USA) was used to measure the content of the cross-linking agent in the synthesised cross-linked PAspNa.

### 2.5. Determination of Molecular Weight of PSI

For molecular weight measurement, LiCl was added to 50 mL of DMF to obtain a concentration of 0.1 M, and 3 g of PSI was dissolved to prepare a PSI solution of concentration 0.06 g/mL. In this case, when LiCl is not added, the concentration dependence of the viscosity shows the curvature normally found in polyelectrolytes of low ionic strength [19]. The viscosity of the PSI was measured using an Ubbelohde viscometer (Shanghai Huichuang Chemical Instrument Co., Ltd., Shanghai, China) in a water bath at 25 °C. ηsp/c and lnηr/c versus c plots were used to obtain the intrinsic viscosities of PSI. The molecular weight of PSI was calculated using the Mark–Houwink–Sakurada (MHS) formula.
(K = 1.32 × 10^−2^ mL/g, α = 0.76)(1)

### 2.6. FT-IR Analysis

FT-IR spectroscopy (PerkinElmer, Spectrum GX FT-IR system, Waltham, MA, USA) was used to observe the structures of L-Asp, PSI, and PAspNa, as well as the mechanism of the cross-linking reaction. After drying the powder sample, it was measured three times to obtain KBr pellets.

### 2.7. DSC Analysis

To investigate the thermal properties of L-Asp, PSI, and PAspNa, the temperature was raised from 30 °C to 300 °C at a rate of 10 °C/min under nitrogen gas using a PerkinElmer DSC-400 equipment, cooled to 30 °C at the same rate, and then increased again. This was measured by reducing the temperature.

### 2.8. Determination of Absorbance by Polyaspartic Acid Sodium Salt (PAspNa)

The absorbance of PAspNa was measured by filtration at room temperature using Advantec Filter Paper No. 2 (Toyo Roshi Kaisha, Ltd., Tokyo, Japan), and both deionised water and 0.9% NaCl solution were used as absorbent solutions. The weight of dried PAspNa was measured and was expressed as W_0_. Then, a test sample of PAspNa was placed in a beaker containing 2000 mL of absorbent solution to absorb water at room temperature for 24 h. The solution in the beaker was poured into a filter funnel, filtered, and left for 15 min. After absorption, the weight of PAspNa was measured and was recorded as *W′*.

The absorbance of PAspNa was calculated as follows:Absorbance of PAspNa (g/g) = *W′*/*W*_0_(2)

## 3. Result and Discussion

We successfully synthesised the PSI by thermal polymerisation with H_3_PO_4_ as a catalyst (Figure 1A). The viscosity of the synthesised PSI was measured using an Ubbelohde viscometer (Figure 2). The intrinsic viscosity of PSI ([*η*] = 27) was obtained from the plots of ηsp/c and lnηr/c versus the concentration (Figure 2). The viscosity average molecular weight of synthesised PSI was calculated to be 23,714 using the Mark–Houwink–Sakurada (MHS) formula as follows:[*η*] = *K*·*M_w_^α^*(3)

In the FT-IR spectrum of L-asp (Figure 3A(a)), a N−H stretching band was observed at 3400 cm^−1^ and the C=O stretching band for the carboxyl group was observed at 1692 cm^−1^. In the FT-IR spectrum of PSI synthesised via thermal polymerisation catalysed by phosphoric acid (Figure 3A(b)), a 1747 cm^−1^ C=O stretching of the imide ring at 3022 cm^−1^ was observed. This showed that the amino and carboxyl groups were converted to an imide ring, confirming the synthesis of PSI [20]. Figure 3B(b) shows the FT-IR spectra of cross-linked PAspNa synthesised using different diamines as cross-linking agents (Figure 1B,C). These spectra show that the 3446 cm^−1^ peak and 1714 cm^−1^ peak of PSI before cross-linking were shifted because the imide ring was converted to carboxylate moiety by hydrolysis. In addition, a strong peak appeared at 3446 cm ^−1^, attributed to the N−H stretching in the cross-linking agent [21].

Figure 3C(a–e) show the FT-IR spectra of cross-linked PAspNa synthesised using diamines of different lengths as cross-linked agents. A cross-linked N-H stretching band at 3446 cm^−1^ was observed for each cross-linked PAspNa. Therefore, the cross-linking reaction was confirmed to proceed well regardless of the length of the cross-linking agent.

DSC has emerged as a powerful physical tool to monitor physical and chemical changes that occur in the polymers during thermal processing and this method yields curves that are unique for synthesised polymers. A DSC analysis of L-Asp (Figure 4A) confirmed that the melting point of L-Asp was approximately 250 °C. The DSC results shown in Figure 4B,C confirm that PSI and PAspNa did not show crystallisation or melting behaviour in the temperature range of 30–300 °C.

PSI was cross-linked at various concentrations to determine the optimal concentration of each cross-linking agent. Among the cross-linking agents, ethylene diamine is a liquid, while the other four cross-linking agents are solids. As shown in Figure 5A, when ethylene diamine was used as the cross-linking agent and the PSI-to-ethylene diamine weight ratio was 1:0.05 and 1:0.07, the synthesised PAspNa was almost entirely water-soluble and did not cross-link. When the PSI-to-ethylene diamine weight ratio was 1:0.09, 1:0.11, 1:0.13, 1:0.15, 1:0.17, and 1:0.19, a cross-linking reaction occurred, such that the synthesised PAspNa was a solid and exhibited water absorption. When the PSI-to-cross-linking agent weight ratio was 1:0.11, the best water absorption was obtained, i.e., the absorptions recorded for deionised water and 0.9% NaCl solution were 45.4 and 20.7 g/g, respectively. Therefore, we can conclude that when ethylene diamine was used as the cross-linking agent at a PSI:ethylene diamine ratio of 1:0.11, with increasing amount of cross-linking agent, the absorbance of synthesised PAspNa first increased and then decreased.

As shown in Figure 5B, when 1,6-hexamethylenediamine was used as the cross-linking agent, and the PSI-to-crosslinking agent weight ratio was 1:0.05 and 1:0.07, the synthesised PAspNa was water-soluble because no cross-linking occurred. When the PSI-to-1,6-hexamethylenediamine weight ratio was 1:0.09, 1:0.11, 1:0.13, 1:0.15, 1:0.17, 1:0.19, PAspNa synthesised by cross-linking showed water absorption. Likewise, when the PSI-to-1,6-hexamethylenediamine weight ratio was 1:0.11, the best water absorption effect was observed, such that the absorbances for deionised water and 0.9% NaCl solution were 156.1 and 50.8 g/g, respectively. In other words, as the amount of 1,6-hexamethylenediamine as cross-linking agent was increased, the absorbance of the synthesised PAspNa first increased and then decreased when PSI:1,6-hexamethylenediamine = 1:0.11.

As shown in Figure 5C, when the weight ratio of PSI to 1,8-diaminooctane as cross-linking agent was 1:0.05 and 1:0.07, the synthesised PAspNa did not cross-link sufficiently to form an emulsion that did not show any absorbency. When the ratio of PSI to 1,8-diaminooctane was 1:0.09, 1:0.11, 1:0.13, 1:0.15, 1:0.17, and 1:0.19, the PAspNa synthesised by cross-linking demonstrated water absorption. The best absorbance was obtained at a PSI-to-1,8-diaminooctane weight ratio of 1:0.11, such that the absorbances for deionised water and 0.9% NaCl solution were 252.6 and 46.5 g/g, respectively. Therefore, we can conclude that when 1,8-diaminooctane was used as the cross-linking agent at PSI:1,8-diaminooctane = 1:0.11, the absorbance of synthesised PAspNa first increased and then decreased with increasing amount of cross-linking agent.

As shown in Figure 5D, when 1,10-diaminodecane was used as the cross-linking agent, the synthesised PAspNa was water-soluble because good cross-linking did not occur when the weight ratios of PSI to 1,10-diaminodecane were 1:0.05, 1:0.07, and 1:0.09. When the ratios of PSI to 1,10-diaminodecane were 1:0.11, 1:0.13, 1:0.15, 1:0.17, and 1:0.19, the PAspNa synthesised by cross-linking showed water absorption. The best absorbance was obtained at PSI:1,10-diaminodecane = 1:0.13, such that the absorbances for deionised water and 0.9% NaCl solution were 197.8 and 43.9 g/g, respectively. Therefore, when 1,10-diaminodecane was used as the cross-linking agent at PSI:1,10-diaminodecane = 1:0.11, the absorbance of synthesised PAspNa first increased and then decreased with increasing amount of cross-linking agent.

In Figure 5E, when 1,12-diaminododecane was used as the cross-linking agent, the synthesised PAspNa was water-soluble because cross-linking did not occur when the weight ratios of PSI to 1,12-diaminododecane were 1:0.05, 1:0.07, and 1:0.09. When the weight ratios of PSI to 1,12-diaminododecane were 1:0.11, 1:0.13, 1:0.15, 1:0.17, and 1:0.19, the PAspNa synthesised by cross-linking showed water absorption. The best absorption was obtained at PSI:1,12-diaminododecane = 1:0.13, such that the absorbances for deionised water and 0.9% NaCl solution were 185.9 and 29.7 g/g, respectively. Therefore, when 1,12-diaminododecane was used as the cross-linking agent at PSI:1,12-diaminododecane = 1:0.11, the absorbance of synthesised PAspNa first increased and then decreased with increasing amount of cross-linking agent.

It was confirmed that the PAspNa salt superabsorbent synthesised using cross-linking agents of different lengths showed a lower absorption in deionised water than the superabsorbents currently in use, but a similar absorption in the salt solution. As shown in the water absorption diagram of all the cross-linking agents, if the diamine cross-linking agent was too short or too long, the water absorption of the synthesised PAspNa was poor. This occurs because of two reasons. If the chain of the cross-linking agent is too short, the distance between the molecular chains during the cross-linking reaction is too small, making postabsorption swelling difficult. In contrast, if the chain of the cross-linking agent is too long, an intramolecular bond can be formed by rotation of the main carbon chain of the cross-linking agent, resulting in low cross-linking efficiency, and a water-soluble product can be produced, which might reduce absorption.

When a small amount of a cross-linking agent is used, the cross-linking reaction does not proceed well, and a network structure is not formed, such that almost all the synthesised PAspNa products are soluble in water. It was challenging to obtain an ideal swelling effect when the moisture-absorbing grid had fewer pores. As shown in Figure 6, the absorbance of PAspNa depends on the length of the cross-linking agent. In general, the absorption increased with an increasing length of the cross-linking agent. However, it can be seen from the graph that the absorption decreases when the main chain of the cross-linking agent contains 10 or more carbon atoms. Table 1 shows the C and N elemental analysis results for PAspNa, and Table 2 shows the estimated introduced cross-linking agent calculated using this method. Table 3 shows the amount of cross-linking agent (mol%) in PAspNa and saltwater absorption under conditions of optimum water absorption. When the cross-linking agent concentration is 0.11 g/g PSI, 1,10-diaminodecane does not have high absorbency owing to the presence of a large number of cross-linking agents. In other words, the higher the content of the cross-linking agent in PAspNa, the lower the absorption of PAspNa.

In this study, an elemental analysis was used to accurately measure the content of the cross-linking agent in PAspNa. However, the cross-linking agent content in PAspNa did not show a regular phenomenon because there appeared to be an error in this experiment; therefore, a detailed analysis could not be performed.

## 4. Conclusions

In this study, the following conclusions were obtained after synthesising a superabsorbent using a diamine cross-linking agent with carbon numbers ranging from 2 to 12 to investigate the effect of the length of the cross-linking agent on the absorbency of PAspNa. The suitable conditions for the cross-linking reaction were determined to be the following: a solvent concentration of 28 mL/g PSI, a reaction temperature of 40 °C, and a cross-linking time of 1 h. The optimum concentration of the cross-linking agent depended on the chain length of the crosslinking agent: the optimum concentration of the cross-linking agent was 0.11 g/g PSI when the main chain of the cross-linking agent contained two to eight C atoms. The absorbency in deionised water was lower than that of conventional superabsorbents, but that in 0.9% NaCl solution reached 50.8 g/g, similar to that of commercially available diapers. When the main chain of the cross-linking agent contained more than 10 C atoms, the optimum concentration of the cross-linking agent was 0.13 g/g PSI, but it was not as effective as the short-chain cross-linking agent. Therefore, with increasing chain length of the cross-linking agent, the effect of the synthesised PAspNa first increased and then decreased. The PAspNa synthesised in this study showed high absorption as well as biogenicity. Since environmental protection is extremely important, the PAspNa superabsorbent can replace the existing nondegradable superabsorbents.

## Figures and Tables

**Figure 1 polymers-14-02244-f001:**
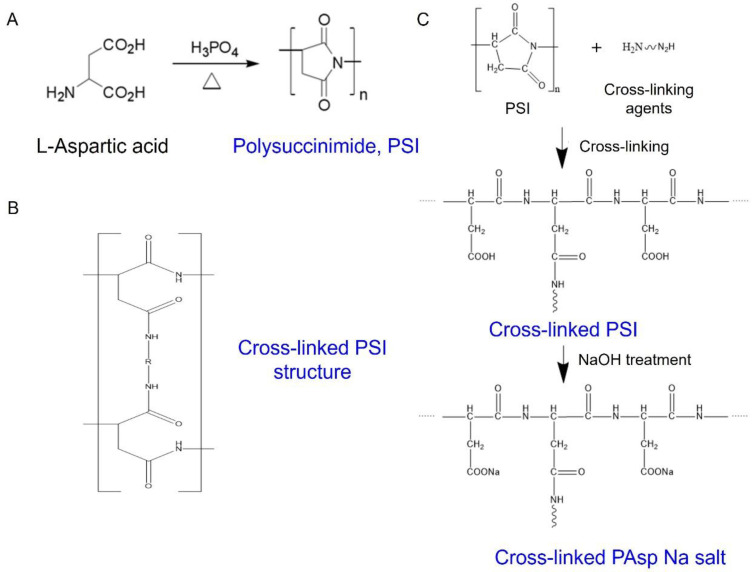
(**A**) Reaction scheme for PSI preparation, (**B**) chemical structure of cross-links in PSI, (**C**) cross-linking of PSI and subsequent hydrolysis to prepare PAspNa.

**Figure 2 polymers-14-02244-f002:**
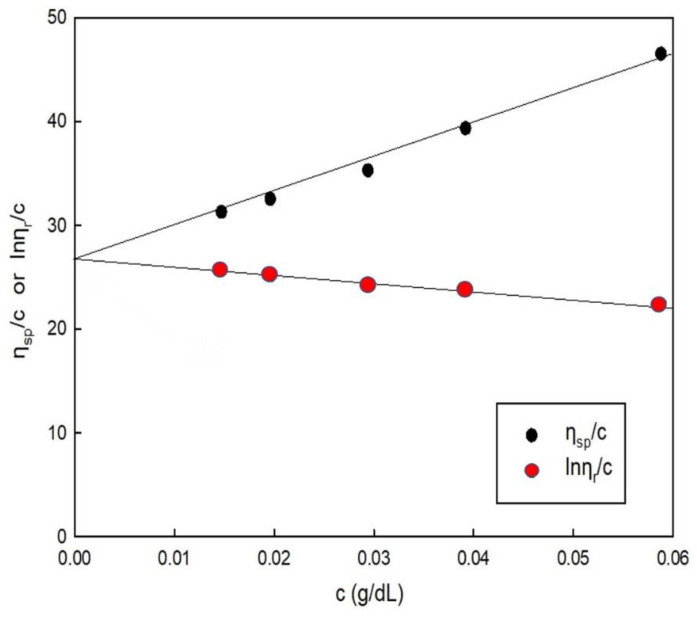
Inherent and reduced viscosity of polysuccinimide measured in DMF (0.1 M LiCl) at 25 °C (average values of three measurements).

**Figure 3 polymers-14-02244-f003:**
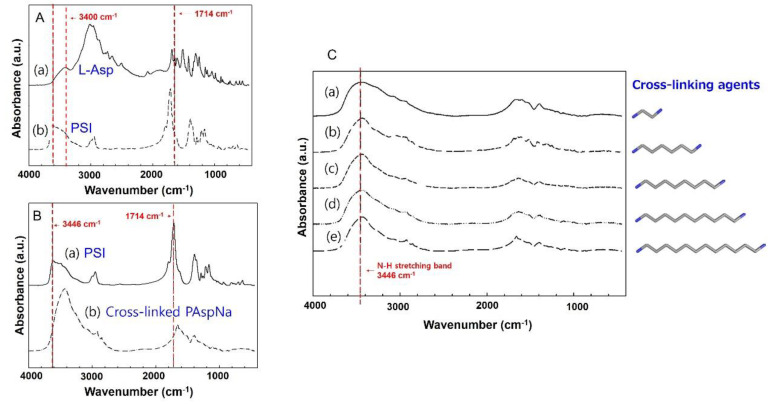
FT-IR spectra of (**A**(a)) L-Asp, (**A**(b)) PSI, (**B**(a)) PSI, (**B**(b)) cross-linked PAspNa samples with different cross-linked agents: (**C**(a)) ethylene diamine, (**C**(b)) 1,6−hexamethylenediamine, (**C**(c)) 1,8−diaminooctane, (**C**(d)) 1,10−diaminodecane, (**C**(e)) 1,12−diaminododecane.

**Figure 4 polymers-14-02244-f004:**
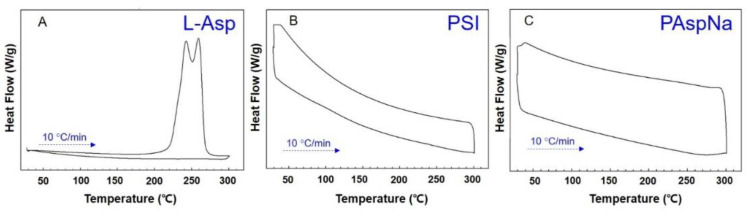
DSC thermograms of (**A**) L-Asp, (**B**) PSI, (**C**) PAspNa.

**Figure 5 polymers-14-02244-f005:**
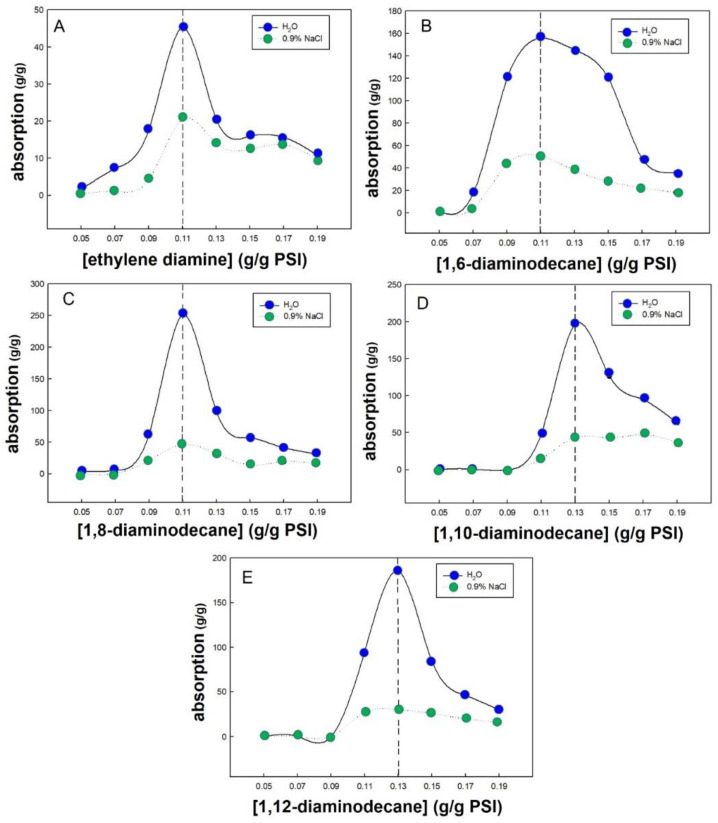
The effect of the concentration of the cross-linking agent: (**A**) ethylene diamine, (**B**) 1,6-diaminodecane, (**C**) 1,8-diaminodecane, (**D**) 1,10-diaminodecane, (**E**) 1,12-diaminododecane, on the water and saltwater absorption.

**Figure 6 polymers-14-02244-f006:**
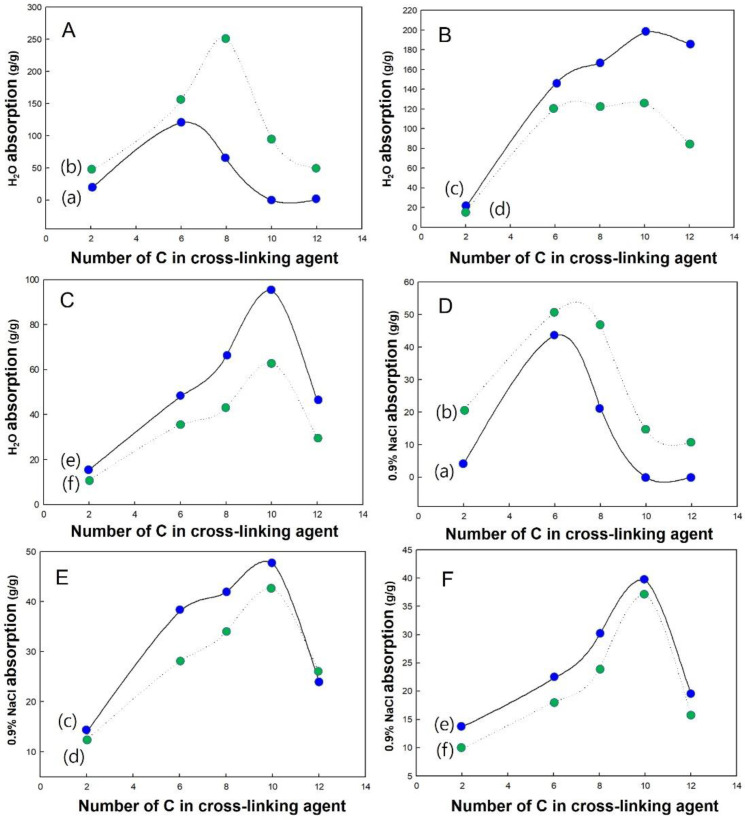
Effect of the number of C in cross-linking agent on the water (**A**–**C**) and salt water (**D**–**F**) absorption: (a) 0.09 g/g PAspNa, (b) 0.11 g/g PAspNa, (c) 0.13 g/g PAspNa, (d) 0.15 g/g PAspNa, (e) 0.17 g/g PAspNa, (f) 0.19 g/g PAspNa.

**Table 1 polymers-14-02244-t001:** C and N contents (wt %) in PAspNa.

Number of C in Cross-Linking Agent	Concentration of the Cross-Linking Agent (g/g PSI)	wt %	
		C	N
6	0.11	29.8	8.6
8	0.11	32.5	9.4
10	0.11	31.2	8.7
12	0.11	31.3	9.1
10	0.13	31.2	9.1

**Table 2 polymers-14-02244-t002:** Amount of cross-linking agent (mol%) in PAspNa and saltwater absorption when the concentration of the cross-linking agent was 0.11 g/g PSI.

Concentration of the Cross-Linking Agent Is 0.11 g/g PSI				
Number of C in cross-linking agent	6	8	10	12
Amount of cross-linking agent (mol%) in PAspNa	0.5	0.5	20	0.5
Absorption of 0.9% NaCl (g/g PSI)	51	47	14.4	30

**Table 3 polymers-14-02244-t003:** Amount of cross-linking agent (mol%) in PAspNa and saltwater absorption under conditions of optimum water absorption.

Best Water Absorption				
Number of C in cross-linking agent	6	8	10	12
Mole% of crosslinking agent in PAspNa	0.5	0.5	0.5	0.5
Absorption of 0.9% NaCl (g/g PSI)	51	47	43.9	30

## Data Availability

The data presented in this study are available on request from the corresponding author.

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
