# Peer review of "Effect of Cross-Linker Length on the Absorption Characteristics of the Sodium Salt of Cross-Linked Polyaspartic Acid"

_polymers, 2022, doi:10.3390/polym14112244_

Round 1

Reviewer 1 Report

The research article “Effect of Crosslinker Length on the Absorption Characteristics of the Sodium Salt of Cross-linked Polyaspartic Acid” by Kyu Oh Kim et al aims to understand the effect of crosslinker length on the behavior of Polyaspartic acid in their efforts for generate Superabsorbent polymer. Authors tested different linkers and confirmed the crosslinking by FT-IR spectroscopy, The degree of crosslinking by elemental analysis and confirmed the thermal properties by dynamic scanning calorimetry. I am enthusiastic about this article and supportive of its publication. I only offer some minor suggestions to improve readability and enhance the message of the paper (adopting them is optional).

Minor issues: 

  1. Error bars missing in Figure 2, 5 and 6. 
  2. Resolution of Fig 5 & 6 could be improved. 
  3. What was the error in elemental analysis experiment authors talked about. If they could explain it, that would be useful for others who could try a similar experiment. 

Author Response

Revision cover letter

Manuscript number: polymers-1691056

Manuscript title: Effect of Crosslinker Length on the Absorption Characteristics of the Sodium Salt of Cross-linked Polyaspartic Acid

Authors: Kyu Oh Kim et al.

We would like to express our thanks for your review and valuable comments. We have tried to answer your questions and have addressed the issues you have pointed out. We believe that the revised manuscript has been improved by reflecting the reviewers’ comments. Answers are marked in blue for better readability. Also, changes in the text are marked in red. We hope the revised manuscript meets the quality standards for publication in polymers.

You will find the attached response to reviewers’ comments next to this letter.

Thank you very much.

Best wishes,

Kyu Oh Kim, Junchao Xu, Kee Jong Yoon

May 6, 2022

Reviewer #1 The research article “Effect of Crosslinker Length on the Absorption Characteristics of the Sodium Salt of Cross-linked Polyaspartic Acid” by Kyu Oh Kim et al aims to understand the effect of crosslinker length on the behavior of Polyaspartic acid in their efforts for generate Superabsorbent polymer. Authors tested different linkers and confirmed the crosslinking by FT-IR spectroscopy, The degree of crosslinking by elemental analysis and confirmed the thermal properties by dynamic scanning calorimetry. I am enthusiastic about this article and supportive of its publication. I only offer some minor suggestions to improve readability and enhance the message of the paper (adopting them is optional).

Minor issues: 

  1. Error bars missing in Figure 2, 5 and 6. 

Response: Thank you for pointing this out. Unfortunately, the error bar cannot be drawn because the measurer only shares the average value.

  1. Resolution of Fig 5 & 6 could be improved. 

Response: Thank you for pointing this out. We have accordingly revised resolution of Fig 5 & 6

  1. What was the error in elemental analysis experiment authors talked about. If they could explain it, that would be useful for others who could try a similar experiment. 

Response: Thank you for pointing this out. Unfortunately, the error bar cannot be drawn because the measurer only shares the average value.

Reviewer 2 Report

In this manuscript, the author reports, ‘Effect of Crosslinker Length on the Absorption Characteristics of the Sodium Salt of Cross-linked Polyaspartic Acid’. The authors should address the following questions before getting a possible publication.

Recommendation: Major revisions needed as noted.

  1. The abstract should be short and precise.
  2. The novelty of the present work should be discussed a little bit more in the Introduction section.
  3. The author should provide high resolution image of Fig3-Fig.6. Also Figure legends are not visible clearly to the readers.
  4. The FTIR results should be properly cited with relevant references.
  5. The formatting and grammatical errors in the article need to be checked carefully. The author should write the purpose for each test in one/two sentences (in brief) before explaining the results of the characterization techniques. Therefore, the logic and organization of this part will be enhanced.
  6. The authors have cited relevant references in the Introduction section; however the manuscript needs to be highlighted further to broaden the impact, related literatures: Acta Biomaterialia, 22, 32-38; ACS Applied Materials & Interfaces, 12(46), 51940-5195 (https://doi.org/10.1021/acsami.0c14527), European Polymer Journal, 59, 363-376; Ultrasonics Sonochemistry, 60, 104797.

Author Response

Revision cover letter

Manuscript number: polymers-1691056

Manuscript title: Effect of Crosslinker Length on the Absorption Characteristics of the Sodium Salt of Cross-linked Polyaspartic Acid

Authors: Kyu Oh Kim et al.

We would like to express our thanks for your review and valuable comments. We have tried to answer your questions and have addressed the issues you have pointed out. We believe that the revised manuscript has been improved by reflecting the reviewers’ comments. Answers are marked in blue for better readability. Also, changes in the text are marked in red. We hope the revised manuscript meets the quality standards for publication in polymers.

You will find the attached response to reviewers’ comments next to this letter.

Thank you very much.

Best wishes,

Kyu Oh Kim, Junchao Xu, Kee Jong Yoon

May 6, 2022

Reviewer #2 In this manuscript, the author reports, ‘Effect of Crosslinker Length on the Absorption Characteristics of the Sodium Salt of Cross-linked Polyaspartic Acid’. The authors should address the following questions before getting a possible publication.

Recommendation: Major revisions needed as noted.

  1. The abstract should be short and precise.

Response: Thank you for pointing this out. We revised the abstract form.

Abstract

Superabsorbent is a polymer material that can absorb water in quantities that are hundreds or thousands of times more than its own weight. Currently, the most widely used superabsorbents are primarily Na salts of cross-linked polyacrylic acid with excellent moisture absorption and moisture retention properties. However, conventional superabsorbents are not biodegradable. Among the previously reported superabsorbents, polyaspartic acid Na salt (PAspNa) and alginate acid Na salt are biodegradable. PAspNa preparation involves polymerising L-aspartic acid to obtain polysuccinimide (PSI), followed by crosslinking of PSI, and subsequent hydrolysis with NaOH to afford PAspNa. In this study, the effect of the crosslinking length on the absorption characteristics of PAspNa was investigated using different concentrations of diamine crosslinking agents bearing carbon chains of different lengths, viz. ethylenediamine, 1,6-hexamethylenediamine, 1,8-diaminooctane, 1,10-diaminodecane, and 1,12-diaminododecane were used as crosslinking agents. The absorption of PAspNa was measured in deionised water and in a 0.9% aqueous NaCl solution. Under the conditions tested, when the alkyl chain of PAspNa was too short or too long, the absorbency was low and the crosslinking length was optimum. The success of the crosslinking reaction was confirmed by FT-IR spectroscopy. The degree of crosslinking was estimated and the ideal concentration for maximum water absorption was determined by elemental analysis. The sample obtained by crosslinking 1,8-diaminooctane at a concentration of 0.11 g/g PSI showed the highest absorption. The thermal properties of each material were determined by dynamic scanning calorimetry (DSC). Therefore, the length of the crosslinking agent was found to strongly influence water absorption.

à revised Abstract

For the development of the biodegradable superabsorbent polymers, the effect of the crosslinking length on the absorption characteristics of The Na salt of polyaspartic acid (PAspNa) was demonstrated using different concentrations of diamine crosslinking agents bearing carbon chains of different lengths, viz. ethylenediamine, 1,6-hexamethylenediamine, 1,8-diaminooctane, 1,10-diaminodecane, and 1,12-diaminododecane were used as crosslinking agents. The absorption of PAspNa was measured in deionized water and in a 0.9% aqueous NaCl solution. Under the conditions tested, when the alkyl chain of PAspNa was too short or too long, the absorbency was low and the crosslinking length was optimum. The success of the crosslinking reaction was confirmed by FT-IR spectroscopy. The degree of crosslinking was estimated and the ideal concentration for maximum water absorption was determined by elemental analysis. The sample obtained by crosslinking 1,8-diaminooctane at a concentration of 0.11 g/g polysuccinimide (PSI) showed the highest absorption. The thermal properties of each material were determined by dynamic scanning calorimetry. Therefore, the length of the crosslinking agent was found to strongly influence water absorption.

  1. The novelty of the present work should be discussed a little bit more in the Introduction section.

Response: Thank you for pointing this out. We have accordingly discussed a little bit more in the Introduction.

Biodegradability and bio-friendly superabsorbent of crosslinked PAspNa argue in favour of the feasibility of future biological applications.

  1. The author should provide high resolution image of Fig3-Fig.6. Also Figure legends are not visible clearly to the readers.

Response: Thank you for pointing this out. We have accordingly revised resolution of Figure 3 – 6

  1. The FTIR results should be properly cited with relevant references.

Response: Thank you for pointing this out. We have properly cited with relevant references.

  1. Yang D, Zhang X, Yuan L, Hu J. PEG-g-poly(aspartamide-co-N,N-dimethylethylenediamino aspartamide): Synthesis, characterization and its application as a drug delivery system. Prog Nat. Sci. 2009;19: 1305–1310.
  2. Zhou Y, Wang J, Fang, Y. Green and High Effective Scale Inhibitor Based on Ring-Opening Graft Modification of Polyaspartic Acid. Catalysts 2021;11: 802-814.

  1. The formatting and grammatical errors in the article need to be checked carefully. The author should write the purpose for each test in one/two sentences (in brief) before explaining the results of the characterization techniques. Therefore, the logic and organization of this part will be enhanced.

Response: Thank you for pointing this out. We have accordingly revised the results and discussion part and added the two sentences.

We were successfully synthesized the PSI by thermal polymerization with H3PO4 as a catalyst. The viscosity of the synthesized PSI was measured using a Ubbelohde viscometer (Figure 2).

DSC has emerged as powerful physical tools to monitor physical and chemical changes that occur in the polymers during thermal processing and these methods yield curves that are unique for synthesized polymers.

  1. The authors have cited relevant references in the Introduction section; however the manuscript needs to be highlighted further to broaden the impact, related literatures: Acta Biomaterialia, 22, 32-38; ACS Applied Materials & Interfaces, 12(46), 51940-5195 (https://doi.org/10.1021/acsami.0c14527), European Polymer Journal, 59, 363-376; Ultrasonics Sonochemistry, 60, 104797.

Response: Thank you for pointing this out. We have properly cited with references in introduction section.

  1. Gyarmati B, Mészár E Z, Kiss L,  Deli M A, László K, Szilágyi A. Supermacroporous chemically cross-linked poly(aspartic acid) hydrogels. Acta Biomater. 2015;22:32-38.
  2. Ganguly S, Das P, Itzhaki E, Hadad E, GedankenA, Margel S. Microwave-Synthesized Polysaccharide-Derived Carbon Dots as Therapeutic Cargoes and Toughening Agents for Elastomeric Gels. ACS Appl. Mater. Interfaces 2020;12:51940-5195.
  3. Sharma S, Dua A, Malik A. Polyaspartic acid based superabsorbent polymers. Eur. Polym. J. 2014;59:363-376.
  4. Ganguly S, Das P, Das T K, Ghosh S Das, Bose M, Mondal M, Das A K, Das N Ch. Acoustic cavitation assisted destratified clay tactoid reinforced in situ elastomer-mimetic semi-IPN hydrogel for catalytic and bactericidal application. Ultrason Sonochem 2020;60:104797.

Round 2

Reviewer 2 Report

The authors have addressed all the questions raised before. Therefore the manuscript can be accepted in its present form.